# Experts’ Opinion for Improving Pertussis Vaccination Rates in Adolescents and Adults: A Call to Action

**DOI:** 10.3390/ijerph19074412

**Published:** 2022-04-06

**Authors:** Giovanni Gabutti, Irene Cetin, Michele Conversano, Claudio Costantino, Paolo Durando, Sandro Giuffrida

**Affiliations:** 1Department of Medical Sciences, Section of Public Health Medicine, University of Ferrara, 44121 Ferrara, Italy; 2Obstetrics and Gynecology, Department “Woman, Mother and Child”—ASST Fatebenefratelli Sacco, Buzzi Children’ Hospital, University of Milan, 20157 Milan, Italy; irene.cetin@unimi.it; 3Department of Prevention, ASL of Taranto, 74121 Taranto, Italy; michele.conversano@asl.taranto.it; 4Department of Health Promotion, Maternal and Child Care, Internal Medicine and Excellence Specialties, University of Palermo, 90133 Palermo, Italy; claudio.costantino01@unipa.it; 5Occupational Medicine, Department of Health Sciences, University of Genoa, 16132 Genoa, Italy; durando@unige.it; 6Occupational Medicine Unit, IRCCS Ospedale Policlinico San Martino, 16132 Genoa, Italy; 7Department of Prevention, Azienda Sanitaria Provinciale of Reggio Calabria, 89124 Reggio Calabria, Italy; sandrogiuffrida@gmail.com

**Keywords:** dTap vaccine, pertussis, pertussis vaccine, pertussis vaccine in pregnancy, pertussis immunization, pertussis immunity duration, pertussis booster for adults

## Abstract

This article highlights the importance of diphtheria-tetanus-acellular pertussis (with reduced antigen content, dTap) vaccination in preventing pertussis, a respiratory infection that is still widespread and easily transmitted. In particular, it highlights the need to receive a booster vaccination throughout life to maintain high antibody levels, which decrease through time. This document collects the opinions that emerged from the comparison between major Italian experts in the field of vaccination. This working group was created to promote a “call to action”, aimed at raising awareness among all institutions, public health authorities, and health workers involved in the vaccination process, about the importance of dTap vaccine administration and with the mindset of implementing the strategic vaccination plan provided by the National Vaccine Plan (NVP). In fact, despite this vaccine being included in the NVP, there are some issues attributable to the practice of vaccination (local health authorities, vaccination centers, occupational health services, gynecology centers, societies of work). Therefore, it is necessary that the Ministry defines the vaccination coverage objectives, identifies the groups of subjects who should receive the booster vaccine (subjects exposed to greater risk of infection, subjects over 60, pregnant women), and applies all the necessary measures to encourage the implementation of this practice.

## 1. Introduction

Pertussis is an extremely contagious infection of the respiratory system, transmitted by the Gram-negative bacterium *Bordetella pertussis* [1,2,3,4,5]. It affects individuals of all ages but has a higher incidence in the pediatric age group and a more significant impact on infants and immunocompromised individuals. In particular, it has been observed that in infants under 1 year of age, 50% of cases are hospitalized and may develop respiratory and neurological complications [3,6,7,8,9,10,11,12,13]. 

Sometimes, in the most serious cases, the disease can lead to death. In Italy, it has been shown that among pertussis hospitalizations, 64% of cases involved subjects less than 1 year old [14]. 

In both adolescents and adults, the severity of the clinical picture is usually lower, especially in subjects who have been previously vaccinated. This last category can represent a source of infection difficult to identify, since the disease can assume a pauci-symptomatic character (mild and often unrecognized symptoms) and may not be correctly diagnosed, also due to a limited use of laboratory diagnostic methods and a reduced number of patients who are actually hospitalized. This is related to an increased risk of transmission in young children who have not yet started or completed the primary vaccination cycle [15], and for whom pertussis continues to be a significant health problem [1,4,16,17,18,19,20,21,22]. 

Infection stimulates both types of the immune response, both humoral and cell-mediated. There is no serological marker that assesses the possible protection from the disease yet; however, it is possible to roughly establish the recent contact with the pathogenic agent, verifying the presence in the blood of antibodies for one or more components of the bacterium. As a matter of fact, high levels of anti-pertussis toxin (anti-PT) are considered indicative of a recent exposure to the pathogen; using ELISA tests, results ≥100 International Units/ml (IU/mL) in adolescents and adults are indicative of recent contact with *B. pertussis* [23,24]. 

In particular, it has been shown that, similarly to what happens for tetanus and diphtheria, natural immunity or immunity acquired with vaccination is not long-lasting and tends to decrease over time: there is a significant reduction after about 10 years, which is more evident and rapid in subjects previously immunized by vaccination compared to those who have developed immunity following the natural disease. For this reason, after 10 years from the first vaccination, it is advisable to carry out periodic vaccination boosters [25,26,27,28,29,30,31,32]. 

The levels of antibodies directed against a single antigen or combination of antigens that can be correlated with certainty to clinical protection are currently unknown.

## 2. How the Epidemiology of the Disease has Changed over Time

National epidemiological evidence underlines that the disease currently involves different age groups than those affected in the past.

Investigations conducted showed that in the pre-vaccination era, the incidence of pertussis was higher in the pediatric group [15,30].

In Italy, the epidemiological picture changed following the recommendation of anti-pertussis vaccination by the Ministry of Health in 1962. Subsequently, the introduction of the acellular vaccine, in combination with diphtheria and tetanus antigens, reduced reported cases. However, a new increase has been recorded since 2016, even in countries with high vaccination coverage, where epidemic episodes [33] have been reported. Moreover, the comparison with other countries suggests that the disease incidence in Italy is significantly underestimated [15]. 

Most cases of infection occur during the first year of life, and a second infectious peak is observed during adolescence: this trend confirms the reduction in previously acquired immune protection. Therefore, adolescents and adults play an important role in the dynamics of disease transmission, as also demonstrated by numerous seroepidemiological studies [34,35], especially in industrialized countries [36]. 

For example, a survey conducted in Sweden showed an increase in the number of cases of whooping cough in the adult population (57% of cases) and the over-50s (17% of cases). In addition, the disease is more frequent in frail individuals who have comorbidities (e.g., Chronic obstructive pulmonary disease: COPD, asthma). 

In 2018, the World Health Organization (WHO) estimated 151,074 cases and 89,000 deaths (in 2008) globally, against a vaccination coverage rate of 86% for the three-dose cycle [37,38]. 

The most recent report from the European Centre for Disease Prevention and Control (ECDC) delineates 35,627 cases of whooping cough in 2018 from 30 countries in the European Union (particularly from Germany, the Netherlands, Norway, Spain, and the United Kingdom). 

The notification rate was 8.2 cases per 100,000 people, slightly lower than in previous years (the 2017 report confirmed 9.4 cases per 100,000 people). Individuals aged ≥15 years accounted for 62% of all reported cases: infants aged <1 year were the most affected age group, with the highest incidence rate (44.4 per 100,000 people and three reported deaths), followed by children aged 10–14 years [15].

Why is whooping cough still circulating, despite the use of the vaccine?

The main reasons, some of which have already been mentioned above, are summarized as follows:There are “disease reservoirs”: these are mainly represented by school-age children, adolescents, and adults, who, coming into contact with the most vulnerable subjects (such as subjects with pathologies and infants under 6 months of age who have not yet completed the vaccination cycle), can transmit the disease.Protective immunity as well as antibodies developed following vaccination against whooping cough are reduced over time, being halved after a short time (about one year) (waning immunity).Low vaccination coverage as well as the evolution of the pathogen towards variants more resistant to the immunity induced by vaccination.The disease often goes undiagnosed, partly due to the lack of laboratory tests.Even though there is an indication for vaccination, citizens feel little involvement in the practice of vaccination [26,29,31,32,39,40,41,42,43,44,45,46,47].

## 3. Whooping Cough Prevention: The Importance of Vaccination and the Current Scenario in Italy

The change that the epidemiology of the disease has undergone over time has influenced the choice of intervention strategies to be implemented to control the transmission of the infection.

The most important preventive strategy is represented by vaccination, whose primary objective identified by the WHO is represented by reducing the risk of developing severe forms of the disease during childhood and achieving a minimum level of vaccination coverage of 90% [29,31].

In 2018, a vaccine program was proposed [48], designed to provide lifelong immunity to the whole population (children, adults, citizens, workers, travelers, and pregnant women). This proposal can be used as a starting model for developing an effective vaccination plan.

As of August 2020, eight countries have adopted several strategies aimed at controlling pertussis, such as implementing maternal immunization programs and vaccine boosters, in individuals over the age of 18 years old [18].

In the National Vaccine Plan (NVP) 2017–2019 (extended until the end of 2021), the Ministry of Health highlighted the usefulness of carrying out boosters against diphtheria, tetanus, pertussis, and poliomyelitis, which maintain the vaccination continuity over time [32] and grant long-term protection in subjects who have been successfully vaccinated or who may be exposed to new risks of contracting the infection and/or disease (e.g., due to the decay of the immune protection provided by vaccination, the reduced possibility of encountering natural boosters, the reintroduction from endemic areas of pathogens now eliminated in our territory).

The document listed the different times during life when it is appropriate to have them and the type of vaccine needed to achieve the goal. This guidance is also present in the latest edition of the Vaccine Calendar for Life (2019), in which booster shots were recommended for all adults [49].

The booster vaccination occurs every 10 years. It can be carried out starting from adolescence, and it is aimed at maintaining the efficacy previously acquired thanks to the administration of the primary cycle carried out in the pediatric age [50], as well as in pregnant women and in subjects in close contact with children or healthcare personnel.

In accordance with the NVP 2017–2021, it is also possible to use, from the age of 4 years, the formulation “adult type” (diphtheria-tetanus-acellular pertussis vaccine with reduced antigen content, dTap), provided that the child’s parents are adequately informed about the importance of booster vaccination in adolescence and that high adolescent vaccination coverage is ensured.

The booster shots that follow the basic vaccination cycle (three doses at 3–5–11/12 months of age) can be interspersed with a variable period of time (5 or 10 years) based on particular situations (e.g., pregnancy, post-exposure prophylaxis, presence of a major wound, travel to highly endemic areas).

Individuals who have not previously received a vaccination do not need to undergo a full vaccination cycle, but they can take a single dose of the vaccine. In fact, in this case, it is highly possible that previous contact with the bacterium may have led to the development of immunological memory [29,31,32].

The booster vaccine involves using the combined vaccine with reduced antigenic content, which also includes the pertussis components. These are purified and inactivated and can stimulate an adequate immune response to hinder the decay of protection observed over time following vaccination.

Currently, in Italy, there are different types of combined vaccines available: diphtheria-tetanus-acellular pertussis-inactivated poliovirus (full antigenic content, DTaP-IPV) or diphtheria-tetanus-acellular pertussis with reduced antigen content-inactivated poliovirus (dTap-IPV, including the polio-specific antigenic component) and dTap (without the IPV component). In relation to the pertussis component, the vaccines available are single-component, three-component, and five-component antigenic vaccines (Appendix A).

Vaccines with an IPV component are recommended up to 11–18 years of age, while dTap vaccines are recommended for 10-year vaccine boosters for adults and pregnant women (prevention of pertussis in the newborn).

## 4. Immunization Strategies

Prevention of pertussis requires an integrated approach. It is necessary to remember that the control of infection can only be obtained by achieving high levels of immunity in all age groups of the population: for this reason, to increase and maintain high levels of vaccination coverage, different immunization strategies have been defined and/or already adopted in several European countries. These include the vaccination of newborns, pre-school children, adolescents and adults, healthcare workers, childcare workers, and pregnant women, as well as the application of the cocoon strategy [51]. 

Vaccine boosters are also indicated for frail people and patients with chronic degenerative diseases, who are at increased risk of developing post-infectious complications [52].

The value of the vaccine recommendation is also linked to the citizens’ acceptance of the vaccine intervention itself. Adherence to this practice is related to several factors, which have been highlighted well in the document “The NVP 2020–2022: recommendations of the National immunization technical advisory group (NITAG) strategic core” [53].

Scientific evidence shows that, given the same immunogenicity and efficacy, better tolerability of the vaccine promotes adherence to the vaccine recommendation. For this reason, the vaccine with reduced antigenic content (dTap-IPV), compared to the vaccine with full antigenic content (DTaP-IPV), has demonstrated better tolerability, with an excellent level of efficacy and immunogenicity. The main general immunization strategies to be implemented are included in Table 1.

## 5. Importance of Vaccination for a Pregnant Woman and Strategies to Increase the Willingness to Become Vaccinated during Pregnancy

Previously, there was a lack of extensive knowledge regarding the usefulness of pertussis vaccination during pregnancy. 

Currently, vaccination of pregnant women is free of charge [31] and is the best strategy to reduce the risk of severe pertussis in infants [16,54].

International public health agencies recommend the administration of the dTap vaccine during gestation [29,32,39,55,56,57,58,59].

In fact, vaccination of a pregnant woman allows the development of an adequate antibody response and its passive transplacental transfer, and during the last trimester of pregnancy, it allows the protection of the product of conception until he or she can receive the scheduled vaccine cycle [60].

The Board of the Calendar for Life highlighted the importance of vaccination, emphasizing the possibility of carrying it out even in conjunction with the administration of the flu vaccine, which should be administered during the second/third gestational trimester, before or during the flu season [32]. 

The Calendar for Life reiterates the need to receive the vaccination during every pregnancy, regardless of a woman’s previous vaccination history or previous infection.

The trivalent vaccine (dTap) should ideally be administered around the 28th gestational week or anytime between the 27th and 36th gestational weeks, as indicated in the National Vaccination Plan 2017–2021 by the Ministry of Health [32]. This allows a pregnant woman to produce enough antibodies.

However, limited knowledge around vaccines, together with the lack of clear recommendations by health professionals, the lack of precise indications at the legislative level, and the difficulty of being immunized during prenatal consultations, has contributed to the poor diffusion of vaccination culture and the failure to achieve an adequate level of vaccination coverage among pregnant women.

Several strategies could be adopted to increase maternal immunization coverage globally.

First, the Ministry should be asked to include any vaccination during pregnancy in the Certificate of Assistance in Childbirth, which would allow obtaining the exact number of vaccinations carried out during pregnancy and, therefore, having a clear picture of the existing vaccination coverage in the territory.

Next, vaccination for whooping cough before the 28th week of gestation should be encouraged, in the case of women at risk of preterm birth, but also in pregnant women who are hospitalized and at risk of premature birth or other pregnancy-related problems: a vaccination would protect the fetus and the newborn in the case of preterm births. This practice is already active in some centers, and its adoption should be encouraged everywhere.

To ensure the total protection of a pregnant woman, strategies that can involve the vaccination of figures who are most in contact with her (e.g., partners) should be selected, giving the possibility to perform the 10-year booster dose of vaccination in case the booster has not been administered in the last decade [61,62].

Vaccination programs during pregnancy involve gynecologists. However, although these specialists are aware of the possibility of whooping cough vaccination in a pregnant woman and recognize its preventive importance, better awareness is needed to encourage women to become vaccinated [32,63], for example, through the creation of “vaccination opportunities” during pregnancy (e.g., the possibility of receiving the vaccination during scheduled ultrasound check-ups at gynecology and obstetrics hospital departments or at gynecologic consulting clinics) [64]. 

To improve vaccination adherence in the territory, it is necessary to adopt a multidisciplinary approach, which can involve not only gynecologists but also other health figures, as highlighted by the recommendations provided by the NITAG expert group, in view of the next NVP 2022–2025. A collaboration between multiple figures, such as gynecologists, midwives, pediatricians, general practitioners, public health specialists, and pharmacists, and the involvement of territorial and occupational medicine, scientific societies, and travel medicine, can promote the use of the vaccine.

To increase the training of health professionals, it could be useful to propose training courses addressed to all health professionals who give support during pregnancy (gynecologists, midwives, general practitioners, public health doctors) [61,64].

A further improvement strategy is represented by the possibility of including gynecological specialists among the group of experts called to develop the guidelines to be included in the National Vaccination Plan.

The creation of educational resources aimed at pregnant women could also improve vaccination acceptance. 

In fact, although pertussis vaccines have been demonstrated to be safe and effective for both mothers and their newborn children, their acceptance is still low [65]. Very low vaccination coverage among pregnant women has been reported in Italy during the last few years [32,66], in contrast to what has been observed in other countries, such as the United Kingdom (UK), where vaccination coverage is consistently above 50%.

For example, in 2018, data submitted by three different Italian hospitals showed a sub-optimal level of coverage (of 4.8%) [67]. The same authors have recently shown that increased rates can be obtained by applying specific educational tools in selected centers of the study [68]. 

However, there is a lack of broader data.

For this reason, it could be useful to include, at the national level, a specific lesson during birthing classes that relates to the recommended vaccinations during pregnancy [64]. 

The effectiveness of transmission of the message in favor of the vaccination intervention also includes a clear and simple presentation of available vaccines. In pregnant women, this should indicate the five-component vaccine, which has solid data supporting its administration. In contrast, the use of the three-component and the single-component vaccine is only supported by, respectively, prospective observational studies and animal-only studies. 

Therefore, the inclusion in the pregnancy diary of a chapter dedicated to vaccinations to be administered during pregnancy could be a useful tool to promote correct information (a strategy already adopted in some centers), as well as the possibility of having an explanatory product sheet, which can be easily consulted in case of need. The strategies to increase maternal immunization are summarized in Table 2.

## 6. Initiatives Taken at the Regional Level to Promote the Immunization of Pregnant Women

At the regional level, several initiatives have been undertaken to increase maternal immunization rates.

In Sicily, a project has been launched to raise awareness of the use of the vaccine in pregnant women, promoted in hospitals at the regional level. This has involved 326 women and has shown that a brief training intervention, implemented during the childbirth classes, has increased the vaccination coverage rate, raising it from 3–7% (in 2018) to higher values in the intervention group (57.7% of women were vaccinated with the diphtheria–tetanus–pertussis vaccine, 47.8% with the flu vaccine, and 64.2% with both vaccines) [69]. It could be useful to extend this intervention to the whole nation using the organizational support of obstetrical teams.

Even the organization of conferences that have involved multiple figures, such as gynecologists, midwives, and pediatricians, has proved to be a useful tool to increase awareness of the vaccine, especially considering the close relationship that these healthcare professionals establish with pregnant women.

For example, in Calabria, following a conference during which the importance of vaccination during pregnancy was discussed, a protocol of collaboration was drawn up between the hospital’s gynecology department and the Provincial Health Agency (Asp). (This has provided the possibility of carrying out the vaccination within the hospital (through the administration of vaccines provided by Asp), even during pre-birth visits.

In the Region of Puglia, a system is currently in place to monitor pregnant women, who, classified as being at risk, are included in a vaccine-registry system. Local initiatives to promote the immunization of pregnant women in some Italian regions are highlighted in Table 3.

## 7. Importance of dTap Vaccination in Adults and Dissemination of Vaccination Culture in Adults and the Elderly Population

The possibility of vaccination against diphtheria–tetanus–pertussis, usually limited to infancy, should be extended to the adult/elderly population, which is also exposed to these diseases, as shown by epidemiological data and by several pieces of evidence. 

Currently, in Europe, diphtheria is considered a disease under control; however, the number of fatal cases in the pediatric population in Spain (2015) and Belgium (2016) has highlighted the need to maintain high protection against this disease through vaccination of all age groups, including adults. 

A very recent European study, which included Italy, showed that in our country, the proportion of individuals unprotected against diphtheria is very high, standing at over 70% in individuals over 65 years of age. The authors of the study identified that the reasons for the alarming result are both the lack of awareness of the value of diphtheria vaccination among the population and the doctors’ failure in monitoring the patients’ vaccination status [15,37]. 

Even though tetanus is a completely preventable disease, a recent seroprevalence study showed that tetanus cases continue to occur in Italy, and the reported notification and hospitalization rates are higher than in other European countries. In most of the identified cases, the subjects were either not or only partially vaccinated. Moreover, in 80% of the cases, the disease occurred in subjects over 64 years of age and with a higher incidence in women than in men [36,70].

With regard to pertussis, the highest prevalence has recently been found in subjects in the 20–29 age group. However, it is also interesting that there is an increase in the number of subjects recently exposed to the disease in the age group >60 years.

It is important to remember that, if contracted in adulthood, this infection can cause comorbidities (pneumonia, urinary incontinence, sinusitis, etc.) and a significant increase in costs for the National Health System [71]. The adult subject represents the main focus of developing an effective vaccination program. In fact, pertussis vaccination, carried out in adulthood, has a double value: directly, it protects vaccinated individuals; indirectly, it ensures the maximum protection to infants (who may come into contact with infected adults).

The Italian NVP recommends the diphtheria–tetanus–pertussis booster for adults every 10 years, which is important mainly nowadays as any preventable infectious disease could complicate the health status of subjects already affected by SARS-CoV-2 [32].

However, dTap vaccination is neglected in adults, as evidenced by the low vaccination coverage rates revealed by several surveys based on specific interviews conducted over time (Progress of the Health Authorities for Health in Italy, PASSI and PASSI d’Argento surveillance systems). 

A recent European study has shown that in Italy, there is still a high proportion of patients over 65 years of age at risk of developing infectious diseases that can be prevented by the administration of a vaccine. Therefore, initiatives should be taken to improve vaccination coverage.

The best strategy to ensure adult/elderly vaccination adherence is represented by the “vaccination habit”, i.e., the activation of a vaccination pathway that begins at birth and continues throughout childhood and adolescence, adulthood, and senescence.

Therefore, the promotion of an active call to vaccination is recommended; this already has been tested in some regions and can be extended to the younger age group of the population (an active call has already been made for the administration of the Human papillomavirus, HPV, vaccine).

Vaccination centers can invite those entitled to vaccination and send written invitations after consulting the registry lists obtained from various internal systems of the Health Authority or from the municipal registry offices. 

It is vital to remember that the lack of active call to vaccination for anyone entitled, such as elderly people, is the denial of a right of the citizen by the National Health System and Regional Health System, as it is part of the new Essential Levels of Care [72].

The levels of vaccination coverage against infectious diseases increased thanks to this strategy, such as for the active call for anti-pneumococcal and anti-herpes zoster vaccination for subjects 65 years old, but also the dTap booster in adult subjects every 10 years, which made it possible to reach a vaccination coverage of over 30%. The COVID-19 emergency had an impact on all health activities, including vaccinations. In particular, there was a particularly significant decrease in coverage rates in adolescents and the elderly [73].

The vaccination message targeted to the adult patient can also be spread through the active involvement of several specialists and healthcare figures (medical specialists, pharmacists, biologists, professional nurses, healthcare assistants) so that the patient can be adequately informed without having to rely on inadequate information channels.

By achieving this, a continuous training/information process must be initiated and extended to all health workers.

An important role is played by the general practitioner (GP), who establishes a special relationship of trust with his or her client/patient. It has been possible to increase the vaccination coverage rate related to the anti-pneumococcal vaccination thanks to the collaboration with this figure [74,75]. 

Nowadays, GPs already administer some types of vaccines, such as the flu vaccine. For example, in Piedmont, GPs administer the anti-pneumococcal and anti-herpes zoster vaccines in subjects over 65 years of age and in patients with heart and lung diseases.

This category of physicians should be more involved in the promotion of vaccinations, wherever they are carried out in vaccination centers, as well as in their administration in their own clinic, and in the identification of those most at risk (diabetic patients, heart patients, those suffering from lung or kidney diseases or rheumatology or oncology, etc.), who represent a category still largely underestimated in the vaccination practice. 

The future aim is to also involve them in the administration of the pertussis booster in adults, as well as in the booking of vaccination appointments (based on age and risk conditions), for example, through the extension of the GP’s software, connected to the adult’s computerized vaccination registry (AVI).

In addition, the possibility of connecting AVI with an app that can be downloaded on a mobile device would give patients the possibility of independently checking their own vaccination status and receiving, via SMS, reminders of any vaccination appointments. By doing this, the GPs could support elderly patients who are less familiar with computer tools.

However, some aspects related to the practice of vaccination are still lacking: among these are the vaccination centers and GPs’ inability to use an effective AVI for adult/elderly patients.

This gap should be filled as soon as possible, as it makes it difficult to obtain data on the number of vaccinations carried out and, therefore, a realistic estimate of the level of vaccination coverage. 

For this reason, thanks to the Italian Alliance for Active Aging (Happy Ageing), the Ministry of Health was requested to obtain a registry vaccine for adults and the elderly to properly assess the actual number of vaccinations administered. By doing this, vaccinated subjects would obtain a notification that can be shared by public health, vaccination centers, general practitioners, and specialists; it is even necessary to receive the Essential Levels of Care funds. 

Finally, this would allow the activation of a computerized vaccination diary/booklet, through which vaccination appointments would be communicated to the patient through notifications. Strategies to improve the vaccination among adults and the elderly are summarized in Table 4.

## 8. Regional Initiatives

It is worth mentioning “Happy Ageing” among the initiatives already undertaken at the regional level to promote the spread of the vaccine message in the adult population, which highlights the importance of diphtheria–tetanus–pertussis vaccination in both adults and the elderly and its implementation due to a proper distribution of the message among the population. 

Trade unions have also worked to disseminate correct information through the organization of regional events focused on supporting vaccination in the elderly population.

In some regions (e.g., Sicily), the vaccination is supported by the Health Service; in fact, the 10-year booster vaccine is reimbursed for the population over 60 years and for those at risk. Hopefully, this facility will also be taken into consideration in other regions.

## 9. The Importance of dTap Vaccination in Workers

The vaccination coverage rates of the entire community can be increased with the implementation of vaccination programs through a close collaboration between occupational health services, vaccination clinics, and the national health system, reducing the number of workers susceptible to vaccine-preventable diseases (VPDs) and at risk of infection in the workplace.

It is well known that exposure to biological agents is a health risk for workers in many professional environments. The Legislative Decree 81/2008, which is the main Italian legislative reference for the management and prevention of biological risk in the workplace, lists the available vaccinations against each biological agent responsible for VPDs [76].

Health monitoring takes place through the implementation of protocols drawn up based on specific risks identified in the workplace. These protocols are periodically updated in accordance with the latest scientific indications and the current regulations. In this context, occupational physicians provide health services aimed at protecting the health of workers exposed to occupational risks, collaborate with employers to identify any work processes that may be a health risk and the most susceptible jobs, provide indications on health surveillance and the advantages/disadvantages of vaccination/non-vaccination, and ensure that vaccinations are carried out based on exposure.

The implementation of vaccination strategies in professional environments can be useful in contributing to the “creation and maintenance of a safe and healthy work environment for all”, as stated in the International Commission on Occupational Health (ICOH), pursuant to Legislative Decree 81/2008 [76,77]. 

With regard to tetanus vaccination, according to Law No. 292 of 5 March 1963, this is mandatory for certain categories of workers (shepherds, farm workers, waste collection/disposal workers) [78], i.e., for those who work in open environments and who are particularly at risk of infection.

The categories of workers for whom specific vaccinations are recommended, including whooping cough vaccination, are identified within the National Vaccine Plan (NVP) 2017–2019, whose validity was extended until the end of 2021 (due to the exceptional conditions caused by the COVID-19 pandemic, by resolution of the State-Regions Conference of 25 March 2021). In particular, the categories of those who are entitled to the pertussis vaccination are represented by [32]:Staff working in neonatal wards;Nursery staff and all other staff who care for infants;Immunocompromised individuals.

Recently, medical and public health institutions, such as the Advisory Committee on Immunization Practices (ACIP), have recommended the administration of a dTap booster dose to healthcare workers (particularly those exposed to newborns, infants, and pregnant women), regardless if they received any previous doses of the vaccine, to reduce the risk of infection and transmission of pertussis [29]. 

Vaccination of healthcare workers in contact with immunocompromised patients, newborns, and infants, as well as pregnant women in the third trimester and mothers after childbirth, referred to as the “cocooning” strategy, prevents the development of the disease in both the workers and the more-susceptible population at high risk of contracting the disease [79,80]. 

## 10. Conclusions

An efficient vaccination system involves detailed planning of the vaccination intervention, high coverage for all age groups, and, as noted above, a reduced risk for infants and young children of contracting severe forms of pertussis. 

To achieve the objectives listed above, it is necessary to recognize the usefulness of receiving vaccination throughout the life course, administering the dTap vaccine early and quickly, and promoting the 10-year booster in different groups of the population and especially in those most susceptible to infection (the elderly, pregnant women, adolescents [81,82], immunocompromised persons [83]). The last category of individuals, having reduced responsiveness to vaccination and/or shorter duration of protection, require special attention from occupational and public health professionals.

The current vaccination system lacks some aspects:There is an absence of an active call for vaccination by the local health authority to administer the dTap vaccine: the anti-pertussis vaccination is included in the Essential Levels of Care, and for this reason, its administration requires an active offer by the local health authority, which is currently absent.There is a lack of adequate citizen-worker involvement.There is insufficient involvement of pregnant women by gynecologists and obstetricians, as well as insufficient information during childbirth classes (vaccination from the 27th to the 36th week of pregnancy).

However, these critical issues could be improved through the implementation of certain strategies:Obtainment of an adequate National Vaccine Database, which would allow the identification of the subjects who have been immunized against whooping cough and the right supply of the vaccine proportional to the number of citizens eligible for vaccination. The current inadequate surveillance and monitoring system for the disease can be improved by obtaining a consensus from scientific societies and institutions.Improved involvement of medical doctors and specialists that regularly care for at-risk individuals, such as gynecologists, obstetricians, general practitioners, and occupational physicians.Definition of guidelines that provide indications on how to actively call for vaccination, especially in relation to adult and elderly patients (for whom regional and national coverage rates, which regulate this aspect, are absent).Compulsory medical certification is required to practice competitive and non-competitive sports activities and attend swimming pools, gyms, fitness centers, sports halls, etc. [29,32].Spreading of vaccination culture, through the spread of the vaccination message to all regions, as suggested in the document “Recommendations of the strategic nucleus of the NITAG Prevention Plan 2020–2022”, to reach vaccination coverage goals, set by the Ministry of Health, and support public health.Better involvement of citizens in vaccination practice, through the implementation of awareness campaigns aimed at adolescents, young adults, and adults [84,85], and through the administration of dTap vaccine within vaccination campaigns or the organization of specific vaccination events within workplaces [29,32,81,86].

For example, in Sicily, the recommendation of the anti-pertussis vaccination was promoted thanks to its inclusion within the annual flu campaign (recommendations for the influenza season 2020/2021) [81,86,87], giving the possibility to the categories expected, by the National Vaccination Plan, to receive the administration of both vaccines by general practitioners or pharmacists.

## Figures and Tables

**Table 1 ijerph-19-04412-t001:** Summing up of the main general immunization strategies to be implemented.

Vaccination of newborns, pre-school children, adolescents, adults, healthcare workers, childcare workers, pregnant women.
“Cocoon Strategy”
Vaccine booster for frail people
Vaccine booster for patients with chronic degenerative diseases
Promote compliance to vaccine pointing out the excellent tolerability of the reduced antigen content vaccine

**Table 2 ijerph-19-04412-t002:** Strategies to increase maternal immunization.

Inclusion in the Certificate of Assistance in Childbirth of any vaccination during pregnancy
Encourage whooping cough vaccination before the 28th week of gestation
Encourage whooping cough vaccination between hospitalized pregnant women and those at risk of premature birth or other pregnancy related problems
Involve all the figures most in contact with pregnant women
Improve awareness among gynecologists to encourage whoopy cough vaccination
Training courses to all health professionals who give support during pregnancy
Include gynecologists in the Experts’ Panel who develop Guidelines of the National Vaccination Plan
Educational courses for pregnant women to improve vaccination acceptance
Inclusion in the pregnant diary of a chapter dedicated to vaccination to be administered during pregnancy

**Table 3 ijerph-19-04412-t003:** Local initiatives to promote the immunization of pregnant women in some Italian regions.

Sicily: Training during the birthing classes Conferences for gynecologists, midwives, pediatricians
Calabria: Protocol of collaboration between hospital gynecology department and Provincial Health Agency
Puglia: Inclusion of pregnant women at risk in a vaccine-registry system

**Table 4 ijerph-19-04412-t004:** Summing-up of the strategies to improve the vaccination between adults and elderly.

Vaccination habit: vaccination pathway throughout all ages
Active call to all people suitable for vaccination (invited by vaccination centers)
Active involvement of several specialists and healthcare figures to adequately inform the patients
Collaboration with General Practitioners (GPs) and occupational physicians for vaccine administration and identification of patients most at risk
Obtain a registry vaccine for adults and elderly by the Ministry of Health

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
