# Peer review of "Experts’ Opinion for Improving Pertussis Vaccination Rates in Adolescents and Adults: A Call to Action"

_ijerph, 2022, doi:10.3390/ijerph19074412_

Round 1
Reviewer 1 Report
Manuscript ID ijerph-1603531
Experts’ opinion for improving dTap vaccination rates: a call to action
This special topic aimed to promote a "call to 28
action", aimed at raising awareness on dTap vaccination rate among general population. Basically there are still most common problems for increasing the coverage of the vaccine.
Major:
- Paragraphs are too fragmented. Please have English editing for more organized and readable.
- Please add figures and tables for enhancing the readers understand the importance of what you want to promote.
- The level of evidence was low as this was only the expert opinion instead of review articles or systematic review.
Author Response
This special topic aimed to promote a "call to 28 action", aimed at raising awareness on dTap vaccination rate among general population. Basically there are still most common problems for increasing the coverage of the vaccine.
Major:
- Paragraphs are too fragmented. Please have English editing for more organized and readable.
Thank you. Please take note that, accordingly to a specific request of the Editor, the paper has undergone English language editing by MDPI (please find enclosed the English Editing Certificate 40192.
- Please add figures and tables for enhancing the readers understand the importance of what you want to promote.
Thank you. We have prepared and included 4 tables. In detail tables have been included in section 4, 5, 6 and 7. Please see lines 196-197, 289-290, 313-315, 416-417
- The level of evidence was low as this was only the expert opinion instead of review articles or systematic review.
Thank you. The fact that this paper has to be considered an Experts’ opinion for improving Tdap vaccination rates has been clearly included in the title of the manuscript. We did not want to make a review or a systematic review on the topic; we rather have made the choice to involve some experts, with different expertise, to have their opinion in an Italian perspective.
Reviewer 2 Report
This revision addresses a topic of interest and impact on public health, however many aspects are lightly addressed and sometimes with statements that seem erroneous. The references are very scarce.
Specific comments
Regarding the title, I suggest to introduce concepts as "pertussis" and "vaccination strategies in the adolescent adult population" more explicitly. I suggest to remove "Tdap", which is a jergon mostly restricted to pertussis experts.
References are missing in the introduction, as in the rest of the manuscript.
For example in the Introduction section, for statements in lines, 55, 56-60, 63-64, 64-66, 68-73, references are required
Regarding the following statement:
".....however, it is possible to roughly establish the recent contact with the pathogenic agent, verifying the presence in the blood of antibodies for one or more components of the bacterium", it is not entirely correct: the levels of antibodies, if the patients were not recently vaccinated, would indicate that the person was in contact with the bacteria but do not give information on whether that contact was recent or ocurred some months ago.
In paragraphs 68-74, the authors should differentiate the duration of the immunity induced by the vaccination with wP vaccine from that induced by the vaccination with aP vaccine. There are many reports showing that the duration of immunity is not the same for both vaccines. Also the authors should differentiate the duration of immunity by natural infection from that induced by vaccination
Lines 78 to 81, please add references. Please clarify whether the statement refer to global epidemiology or to Italy.
Regarding the pertussis epidemiology in Italy, for a better understanding it would be interesting that the authors introduce a brief description of the surveillance system, graphs of cases per year and vaccination schedule variation over time.
Regarding the causes put forward in paragraphs 109 to 120, references are once again absent. The paragraph does not clearly include the reasons that have more consensus such as waining immunity (protective immunity not only of antibodies), low vaccination coverage, improvements in diagnosis and surveillance strategies and the evolution of the pathogen towards variants more resistant to the immunity induced by vaccination
Paragraph 126-129, need references
Paragraph 160-163, need references
line 183, vaccination of newborns? This strategy does not yet exist, please clarify
lines 192-196, although it is correct, the authors should deepen this concept because, for the pediatric population, there are two types of vaccines (wP and aP) that do not induce the same immune response profile. The authors should describe that wP vaccine is not recommended for the adult and adolescent populations due to adverse reactions.
lines 206-207, does it refer to Italy?
line 208-209, add references
209-212, add more references
223-239, differentiate recommendations for Italy from possible global recommendations
Expand the concepts of paragraph 240-243.
Section 5 would have to be ordered, it comes and goes and repeat concepts
In section 6, it would be good to promote the study of the effect of immunization during pregnancy, for example, in reducing the lethality of babies. Undoubtedly, these data will favor the expansion of the use of vaccination during pregnancy.
lines 315-317, add references
lines 366-368, what happened to vaccination coverage during the pandemic?. It would be interesting to add comments about it.
Although the last sections provide an interesting overview, the manuscript requires a thorough review before it is considered for publication.
Author Response
This revision addresses a topic of interest and impact on public health, however many aspects are lightly addressed and sometimes with statements that seem erroneous. The references are very scarce.
Specific comments
Regarding the title, I suggest to introduce concepts as "pertussis" and "vaccination strategies in the adolescent adult population" more explicitly. I suggest to remove "Tdap", which is a jargon mostly restricted to pertussis experts.
Thank you. The title has been changed as follows: Experts’ opinion for improving pertussis vaccination rates in adolescents and adults: a call to action
- References are missing in the introduction, as in the rest of the manuscript.
For example in the Introduction section, for statements in lines, 55, 56-60, 63-64, 64-66, 68-73, references are required
Thank you. The required references have been included.
- Regarding the following statement:
".....however, it is possible to roughly establish the recent contact with the pathogenic agent, verifying the presence in the blood of antibodies for one or more components of the bacterium", it is not entirely correct: the levels of antibodies, if the patients were not recently vaccinated, would indicate that the person was in contact with the bacteria but do not give information on whether that contact was recent or occurred some months ago.
Thank you. We do not agree with this statement as high levels of anti-PT are considered indicative of a very recent exposure to pertussis. We have clarified this point and added some references.
- In paragraphs 68-74, the authors should differentiate the duration of the immunity induced by the vaccination with wP vaccine from that induced by the vaccination with aP vaccine. There are many reports showing that the duration of immunity is not the same for both vaccines. Also the authors should differentiate the duration of immunity by natural infection from that induced by vaccination
Thank you. The comment is very correct. However, waning immunity is a fact and, in the perspective of our paper, it seems not strictly necessary to detail too much this point. Whatever the vaccine used or natural infection, the need for boosters remains a priority. For these reasons we would like to not modify this part.
- Lines 78 to 81, please add references. Please clarify whether the statement refer to global epidemiology or to Italy.
Thank you. The statement refers to both global and Italian epidemiology. Two references have been added.
- Regarding the pertussis epidemiology in Italy, for a better understanding it would be interesting that the authors introduce a brief description of the surveillance system, graphs of cases per year and vaccination schedule variation over time.
Thank you. The epidemiological trend in Italy, as well as in other countries, has been already included in the paper (lines 115-126 of the original version). References have already been included as well. For these reasons, we would prefer to not modify this part; anyway, we have added another reference (ECDC, 2020)
- Regarding the causes put forward in paragraphs 109 to 120, references are once again absent. The paragraph does not clearly include the reasons that have more consensus such as waning immunity (protective immunity not only of antibodies), low vaccination coverage, improvements in diagnosis and surveillance strategies and the evolution of the pathogen towards variants more resistant to the immunity induced by vaccination
Thank you. We have added some references
- Paragraph 126-129, need references
Paragraph 160-163, need references
Thank you. Accordingly to this request, some references have been added
- line 183, vaccination of newborns? This strategy does not yet exist, please clarify
Thank you. Vaccination of newborns means that at 2-3 months of age, accordingly to each country and its vaccinal schedule, any newborn is vaccinated against pertussis. For example, in Italy any newborn at 3 month of age starts his immunization schedule with the first dose of hexavalent vaccine
- lines 192-196, although it is correct, the authors should deepen this concept because, for the pediatric population, there are two types of vaccines (wP and aP) that do not induce the same immune response profile. The authors should describe that wP vaccine is not recommended for the adult and adolescent populations due to adverse reactions.
Thank you. We have changed the title of the paper including more explicitly adolescents and adults. It seems not strictly necessary to deepen the differences between wP and aP for the pediatric population nor to add anything about wP, that is no more used, since many decades, in Italy as well as in many other countries
- lines 206-207, does it refer to Italy?
Thank you. We have clarified that in Italy vaccination for pregnant women is free of charge including a specific reference
- line 208-209, add references
- 209-212, add more references
Thank you. We have added some references
- 223-239, differentiate recommendations for Italy from possible global recommendations
Thank you. This part follows a number of statements related to Italy and we would like to not change this part.
- Expand the concepts of paragraph 240-243
Thank you. We have included references on this part and would not add anything
- Section 5 would have to be ordered, it comes and goes and repeat concepts
Thank you. As other Reviewers have not pointed out this, we would prefer to not change this part
- In section 6, it would be good to promote the study of the effect of immunization during pregnancy, for example, in reducing the lethality of babies. Undoubtedly, these data will favor the expansion of the use of vaccination during pregnancy.
Thank you. It would probably be quite difficult to promote vaccination during pregnancy using mortality data; probably it would be better to refer to hospital discharge forms that clearly show the impact of pertussis in newborns. As this part has already been included explaining the epidemiological trend of pertussis in other parts of the paper, we would like to not add anything at this point.
- lines 315-317, add references
Thank you. We have added some references
- lines 366-368, what happened to vaccination coverage during the pandemic?. It would be interesting to add comments about it.
Thank you. We have added a sentence as well as a reference on this specific point
- Although the last sections provide an interesting overview, the manuscript requires a thorough review before it is considered for publication.
Thank you. We have changed the paper accordingly to Reviewers’ comments, hoping to have made the manuscript more clear and complete.
Reviewer 3 Report
Dear Authors
First of all, I would like to congratulate you on the paper you have presented. From my point of view this paper is not a review. It could be a narrative review. As you know, a Narrative Review is done by experts and talks about general issues. The issue addressed in this paper may be more or less general but the approach to the problem is not. Moreover, you state that you are a group to promote the call to action. So with all that said, I believe that this paper is not a review but a letter to the editor. My suggestion in this regard is to seek many more signatories with expertise in public health, gynaecology, and paediatrics to support the manifesto.
On the other hand, I support some of the statements made in this paper but have some comments on the text:
On lines 64-66 it is stated that there is no serological marker, but I have doubts about the use of ELISA and other tests to determine the immune status against these diseases. From what I understand, some hospitals are starting to use this technique before injecting booster doses and thus personalise the vaccination.
I recommend that some of the statements be supported by references. For example the statement on line 80 and the Swedish study on line 94, or the one on line 100 referring to the study [2] (but it has to be mentioned) or the one starting on line 126, etc.
Line 185: "frail people" is a term I would avoid using.
Line 188: The paper [15] highlights only one factor associated with adherence to vaccination: that “health organisations give a single message and do not create confusion”. But there are many more factors, such as the way citizens inform themselves or the political leanings of citizens, to give two examples. The problem is much more complex than what is reflected in document [15] and it is for this reason that adherence to vaccines is one of the top health problems on the planet according to the WHO, it is not just an administrative issue.
In line 223 you state "limited knowledge around vaccines together ...". This lack of knowledge is by whom? Please specify.
Finally, after reading the recommendations, some of them are already implemented in other countries and the content is very much contextualized in Italy. In this respect, either change the title of the paper or broaden the framework by extending it to third countries.
Congratulations again.
Author Response
First of all, I would like to congratulate you on the paper you have presented. From my point of view this paper is not a review. It could be a narrative review. As you know, a Narrative Review is done by experts and talks about general issues. The issue addressed in this paper may be more or less general but the approach to the problem is not. Moreover, you state that you are a group to promote the call to action. So with all that said, I believe that this paper is not a review but a letter to the editor. My suggestion in this regard is to seek many more signatories with expertise in public health, gynaecology, and paediatrics to support the manifesto.
Thank you. We agree that this is not a review. However, Experts involved in preparing this document well represent the expertise requested being mostly involved in public health and other specialties very relevant in Italy in supporting immunizations and vaccinal strategies as well.
On the other hand, I support some of the statements made in this paper but have some comments on the text:
On lines 64-66 it is stated that there is no serological marker, but I have doubts about the use of ELISA and other tests to determine the immune status against these diseases. From what I understand, some hospitals are starting to use this technique before injecting booster doses and thus personalise the vaccination.
Thank you. We have changed some part of the paper in order to make more clear the relevance of serological results in a general perspective. It is absolutely correct that no serological marker is available for pertussis. On the other hand, it is well known that high levels of anti-pertussis toxin are considered indicative of a very recent contact with B. pertussis. In Italy no personalization of immunization is performed and serological test are not used before administering a booster dose.
I recommend that some of the statements be supported by references. For example the statement on line 80 and the Swedish study on line 94, or the one on line 100 referring to the study [2] (but it has to be mentioned) or the one starting on line 126, etc.
Thank you. Accordingly to these recommendation as well as the ones done by other Referees, we have included several references supporting more clearly the different statements of the document.
Line 185: "frail people" is a term I would avoid using.
Thank you. With the term “frail people” we consider subjects that because of age and/or co-morbidities and/or immunosenescence are at high risk of complications due to infections. We would prefer to not change this term
Line 188: The paper [15] highlights only one factor associated with adherence to vaccination: that “health organisations give a single message and do not create confusion”. But there are many more factors, such as the way citizens inform themselves or the political leanings of citizens, to give two examples. The problem is much more complex than what is reflected in document [15] and it is for this reason that adherence to vaccines is one of the top health problems on the planet according to the WHO, it is not just an administrative issue.
Thank you. Absolutely correct. The document included as a reference points out the complexity of this topic
In line 223 you state "limited knowledge around vaccines together ...". This lack of knowledge is by whom? Please specify.
Thank you. This part refers to vaccination in pregnant women as it is clearly stated at the end of the sentence. For this reason, we would like to not modify this part
Finally, after reading the recommendations, some of them are already implemented in other countries and the content is very much contextualized in Italy. In this respect, either change the title of the paper or broaden the framework by extending it to third countries.
Thank you. Accordingly to this point, we have changed the title of our paper
Congratulations again.
Thank you for this
Round 2
Reviewer 1 Report
More concise English writing would be appreciated as there are lots fragmented paragraphs. Otherwise, I have no further comment.
Reviewer 2 Report
The authors have improved the manuscript; however, the following comments should be addressed before the manuscript is considered for publication.
The comments that I prevously performed for paragraph in lines 109 -118 (previous version , current version, lines 112-121) were not addressed
...
The paragraph does not clearly include the reasons that have the most consensus, such as decreased immunity (protective immunity not only of antibodies), low vaccination coverage, improvements in diagnostic and surveillance strategies, and the evolution of the pathogen towards variants more resistant to infection.
It would also be important to indicate, for lines 173 and 174, the composition and brand of the indicated vaccines available (single-component, three-component and five-component antigenic vaccines).
Paragraph 180-183 is not correct, vaccines do not interrupt transmission, in any case they reduce it a little. Please modify accordingly
Regarding the term newborn, this term generally refers to a baby from birth to about 2 months of age. The term infants can be considered children from birth to one year of age, I suggest using this distinction for clarity.
